# Using Digital Twin Documents to Control a Smart Factory: Simulation Approach with ROS, Gazebo, and Twinbase

Joel Mattila *[ID], Riku Ala-Laurinaho [ID], Juuso Autiosalo [ID], Pauli Salminen [ID] and Kari Tammi [ID]

Department of Mechanical Engineering, Aalto University, 02150 Espoo, Finland;
riku.ala-laurinaho@aalto.fi (R.A.-L.); juuso.autiosalo@aalto.fi (J.A.); pauli.salminen@aalto.fi (P.S.);
kari.tammi@aalto.fi (K.T.)
* Correspondence: joel.mattila@aalto.fi

**Abstract:** Digital twin documents are expected to form a global network of digital twins, a "Digital Twin Web", that allows the discovery and linking of digital twins with an approach similar to the World Wide Web. Digital twin documents can be used to describe various aspects of machines and their twins, such as physical properties, nameplate information, and communication interfaces. Digital twin is also one of the core concepts of the fourth industrial revolution, aiming to make factories more efficient through optimized control methods and seamless information flow, rendering them "smart factories". In this paper, we investigate how to utilize digital twin documents in smart factory communication. We implemented a proof-of-concept simulation model of a smart factory that allowed simulating three different control methods: centralized client-server, decentralized client-server, and decentralized peer-to-peer. Digital twin documents were used to store the necessary information for these control methods. We used Twinbase, an open-source server software, to host the digital twin documents. Our analysis showed that decentralized peer-to-peer control was most suitable for a smart factory because it allowed implementing the most advanced cooperation between machines while still being scalable. The utilization of Twinbase allowed straightforward removal, addition, and modification of entities in the factory.

**Keywords:** digital twin; smart factory; simulation; robot operating system; ROS; machine-to-machine communication; architecture; client-server; peer-to-peer; P2P

## 1. Introduction

Smart factories are often seen as factories that can quickly adapt to the customers' needs by offering customized production [1], detailed information of production times [2], and conditions of the supply chain. Smart factories are designed digitally, and, thus, their performance can be simulated [3] and evaluated before building actual plants [4]. In the 2010s, one of the megatrends was Industry 4.0, which connects smart factories to networks and cloud-based services [5,6]. These services can use big data, data mining, and artificial intelligence for many purposes, such as predictive maintenance [7], anomaly detection [8], and other monitoring purposes. As smart factories are moving more and more towards complicated cyber-physical systems [9], there is an increasing need for better structural ways to build these systems.

Smart factories consist of machines that communicate with each other. In contrast to legacy factories where machines need to be set up manually for each purpose, the machines of smart factories adapt to their tasks automatically. Human intervention is only needed for innovative tasks, such as designing new machines and communication procedures, and problem-solving in case of faulty operations or emergency situations emerging in the factory. Mundane tasks such as setting up machines are becoming increasingly automated.

This paradigm change is driven by globally increasing labor costs and requirements for more flexible production. It is enabled by autonomous mobile robots and delivery

vehicles. Autonomous machines need to be able to exchange information even if they have never met before, similarly to people. At the beginning of the digital manufacturing era, the factories were operated by a centralized computer system or by using simple programmable logic controllers. Then came the Ethernet-based systems, which were followed by cloud-based systems. Cloud-based systems made more decentralized systems and services available [10,11].

Today, more and more machines, devices, and products have DTs (digital twins) [12] that can be connected to smart factories and other systems [13]. DTs are seen as an essential piece of smart factory infrastructure [14]. In the engineering domain, DTs have been used as sophisticated simulation models connected to real data. These kinds of DTs help monitor machines and predict their maintenance needs. To arrange communication, other types of DTs are needed. For these information management-oriented DTs [15] standardization is crucial so that they can understand each other: DTs need to be formatted in similar methods and talk the same language. An information management-oriented DT aims to provide access to the information stored in the DT.

To allow information flow between machines, their DTs need to talk the same language. Machine-readable meanings of words can be communicated globally through the use of vocabularies, such as SAREF [16], Schema.org [17], and GS1 Vocabulary [18]. (For the purposes of this article, vocabulary is the same as ontology.) These vocabularies are formatted in JSON-LD and other Linked Data formats. We use the phrase "DT document" (digital twin document) to refer to the general concept of a document written to describe a DT. In a previous study, we created a draft specification for DT documents [19], but now it seems that the general concept of a DT document is a more important outcome than the presented specification. Nevertheless, the draft served as a good communication method for concretizing the final goal, and the basic principles described in the paper are still valid. There are also other specifications for writing DT documents such as Web of Things Thing Description by World Wide Web Consortium [20], Digital Twins Definition Language by Microsoft Azure [21], and Asset Administration Shell by Plattform Industrie 4.0 [22]. In this paper, we use our own DT document specification to be able to concentrate on concepts instead of the format. In the long run, we aim to merge the different methods into one general approach. To be useful, this general approach must achieve the position of a standard or de facto standard. In the authors' opinion, DT document specification should be developed as a collaborative effort.

Digital Twin Web (DTW) is a network of digital twins formed by DT documents that describes the contents of DTs and the relationships between the DTs [23]. This type of network of interlinked DTs seems to be the next phase in the development of DTs. DTW is analogous to the World Wide Web but consists of digital twins described in DT documents instead of web pages described in hypertext markup language. As digital twins are counterparts of real-world entities describing their properties and data, DTW starts to mirror the real world and enable the discovery of its information interfaces. When such a network has been built, it can be used in several ways. For example, DT documents can provide access to product information from supply chains, manufacturing, and maintenance to employees. This can be used to connect smart factories directly to products and to alert factory workers when new products of spare parts are needed.

Twinbase is a server solution for managing and distributing DT documents developed by the authors [24]. Twinbase is a combination of a traditional static web server connected to a Git repository and additional custom features for managing DT documents. It can also be hosted free-of-charge at GitHub Pages. In the current study, we used Twinbase to store DT documents that define the basic properties of a simulated factory and entities of interest in that factory. The source code of this Twinbase implementation and the DT documents are publicly available at https://github.com/Zoelz/twinbase (accessed on 15 February 2022). In addition, the user interface for this Twinbase is available at https://zoelz.github.io/twinbase/ (accessed on 15 February 2022).

DT document-based M2M communication differs from traditional machine-to-machine (M2M) communication by adding a metadata layer on top of the communication. In traditional M2M communication, each connection needs to be added manually by technicians based on human-readable documentation. There are ways for devices to find other devices using systems implemented within communication protocols, like inquiry messages used in Bluetooth. The device will send the inquiry message in different frequencies so that other devices that listen to those frequencies can contact it [25]. There are also build systems such as the LWM2M Meta object, which contains a location for an XML file containing information about the object [26].

Using DT documents, machines can initiate new connections automatically based on the standardized interface descriptions found in the DT documents. This proposed method of using DT documents fills a gap of making M2M communications more scalable and automated. The proposed method is investigated with well-known simulation tools and a DT document distribution solution. The proposed method seems to enable fundamentally novel architecture for factory M2M communication.

This paper investigates how the early versions of DTW and Twinbase could be used to control a smart factory, as shown in Figure 1. We implemented a simulation model to compare three types of control methods: centralized client-server, decentralized client-server, and decentralized peer-to-peer. The source files of the simulation model can be accessed from https://github.com/Zoelz/simulationModelControl (accessed on 15 February 2022).

The main contributions of the paper are as follows:

1. Method for using DT documents to enable machine-to-machine communication in smart factory;
2. Implementation of a simulation environment of DT document-based M2M communications in factory with open-source software (Robot Operating System (ROS), Gazebo, Python, and Twinbase);
3. Comparison of three different architectures for controlling a factory.

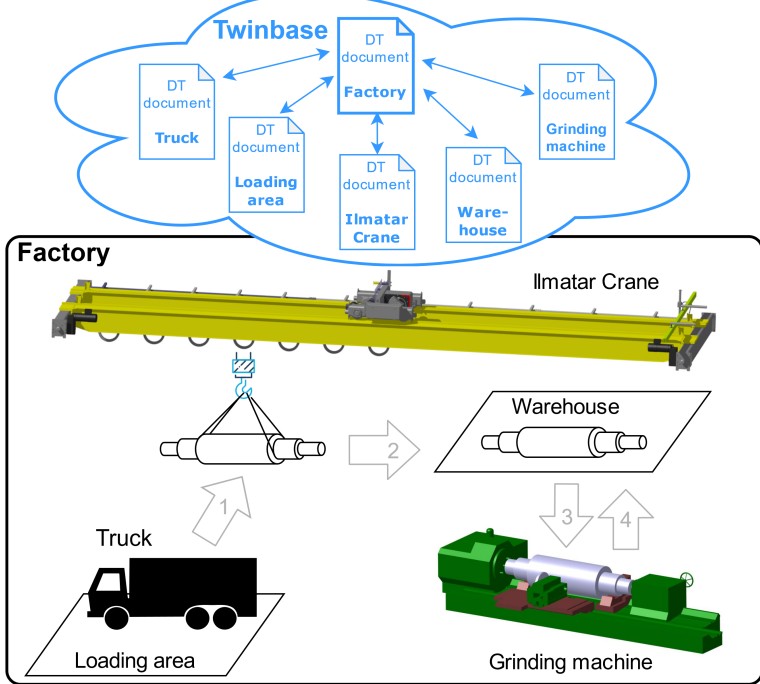

**Figure 1.** Overview of the factory operation and its DT documents. The gray arrows 1–4 show the operating logic of the factory, i.e., unloading rolls, storing them in a warehouse, and maintaining them on a grinding machine. Each object and the factory itself have corresponding DT documents that are stored in Twinbase. The DT documents have parent–child relationships to enable the discovery of other machines located in the factory.

## 2. Methods

In order to test the general functionality of factory control and communication, we established a high-level mission to move material in the factory. As the first step, our aim was to receive an unfinished roll from the truck in the loading area. The second step was for the crane to pick up the roll in the loading area and move it to the grinding machine. The roll was ground and transported to the warehouse in the third step. In the final step, the finished roll was made available for further use in the warehouse.

To examine the different control methods of a smart factory, we implemented a simulation model using open-source software. The simulation model can be divided into operation environment and control implementations. We implemented three types of communication methods: centralized client-server, decentralized client-server, and decentralized P2P (peer-to-peer). In the centralized implementation, all communication went through the same central node compared to decentralized implementation with no central node. In this paper, we use the term "system" when we refer to the whole simulation model, including the operating environment and a certain control implementation.

The control implementations followed either a client-server or P2P model. In a client-server model, nodes are either clients or servers. Client nodes contact servers that respond to clients, thus, all communication is initiated by clients. In P2P implementations, all the nodes are peers that can act as both clients and servers. P2P systems are described by Milojicic et al. [27] as follows: "As a mind set, P2P is a system and/or application that either (1) takes advantage of resources at the edge of the system or (2) supports direct interaction among its users."

Client-server and P2P models use the request-response communication method. Request-response is a communication method in which one node sends a request to another node that responds to it. An alternative for this method is the publish-subscribe model, in which publishers send data to the event manager from which subscribers can then request the data they want. The space decoupling, in which publishers do not know who is subscribing for the data and the subscribers who are publishing the data, can be considered one of the main strengths of the publish-subscribe model [28]. However, the request-response method is more suited for control implementations than the publish-subscribe method because the arrival of a message is acknowledged by the sender [29]. Therefore, we judged the request-response method as being more reliable than the publish-subscribe method and used it in control implementations. However, it is important to note that ROS, which was utilized to build the simulation model, uses the publish-subscribe method in its internal communication.

### 2.1. Operating Environment

The operating environment represented the physical counterpart of the system. This physical counterpart mimicked a part of a smart factory and it consisted of a crane, a grinding machine, and a truck. The operating environment was created using Gazebo, *Gazebo OPC UA server bridge*, and OPC UA servers, as seen in Figure 2. The operating environment was not modified when the control implementation of the system was changed.

Gazebo was used as the physics engine and to visualize smart factory operations. Simplified 3D models for machines were created to represent those machines in Gazebo. ROS and "differential_drive_controller" package [30] were applied to control the machines. ROS publish nodes were used to publish commands for the "differential_drive_controller" package and subscribe nodes to get data from machines in a Gazebo environment. There were specified areas in the Gazebo environment: a warehouse, a loading area, and a factory. These areas did not have 3D models, but instead, the control server and DT document represented the warehouse in the simulations. The loading area and the factory only had DT documents as they were merely static objects represented with fixed data. On the contrary, the warehouse also included dynamic data: other entities could ask whether the warehouse has space in it.

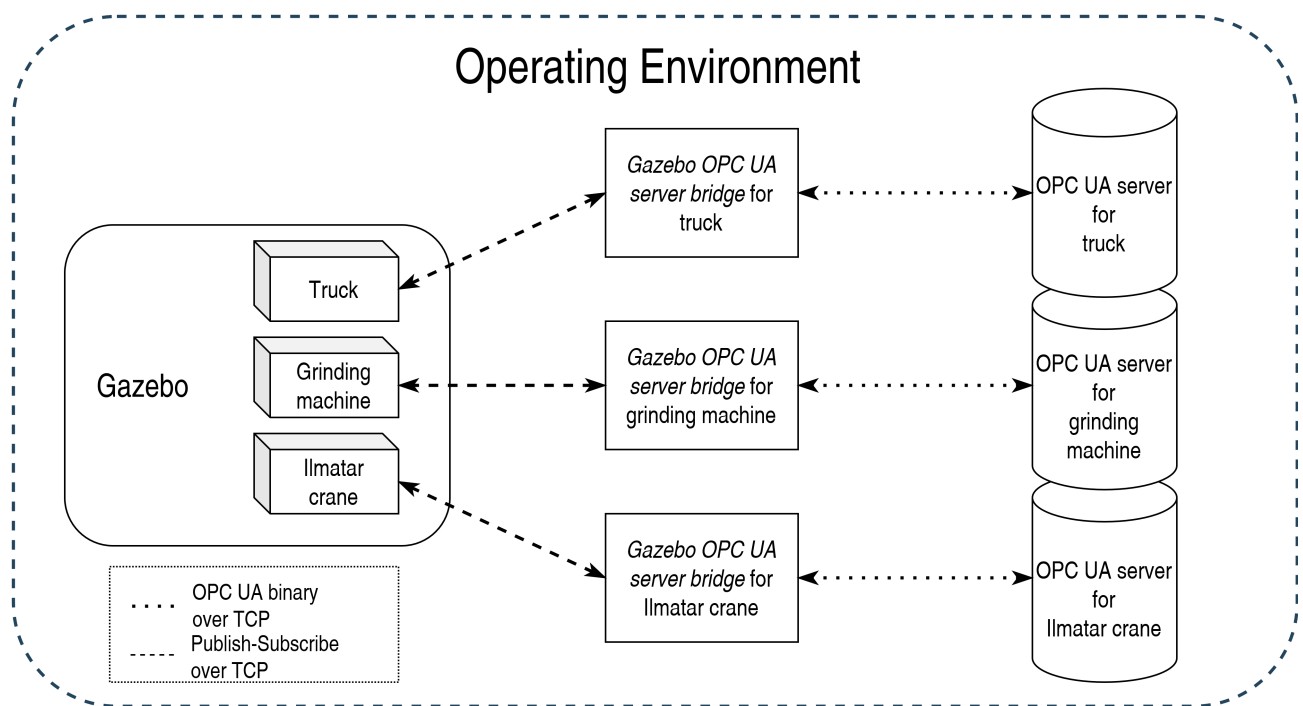

**Figure 2.** OPC UA servers act as an API for the system in the operating environment, and Gazebo is a physics engine and visualization for the system. *Gazebo OPC UA server bridges* transfer data between an OPC UA server and Gazebo. The arrows indicate that the communication can be initiated and data can flow in both ways.

OPC UA servers functioned as an API for communicating with machines from outside of the operating environment. These servers allowed both giving commands to machines and getting information from them. The OPC UA server of the crane followed the same structure as its real-world counterpart, Ilmatar crane [31] OPC UA server. However, OPC UA servers for other machines were not based on existing servers.

When machines contacted other machines, they contacted their control servers rather than their OPC UA servers, as opening up OPC UA verification each time would have taken eight messages before any data can be read or written [32]. Another reason to prefer control server communication was the opportunity to run higher-level functions based on given commands without adding extra nodes to the OPC UA servers. As an example of a higher-level command for a machine, we could ask the crane to drive next to the truck, which also requires finding the (constantly changing) location of the truck.

*Gazebo OPC UA server bridge* enabled communication between the OPC UA server of a machine and Gazebo environment in Figure 2. These *Gazebo OPC UA server bridges* represented PLC logic of the machines for OPC UA servers. *Gazebo OPC UA server bridge* communicated with Gazebo using ROS publish and subscribe nodes. OPC UA binary over TCP was utilized for communicating with OPC UA servers.

### 2.2. Controlling Implementations

In this paper, three control implementations were created: centralized client-server, decentralized client-server, and decentralized P2P, defined at the beginning of this chapter. All the control implementations were connected to the operating environment, including Gazebo and OPC UA servers in Figure 3. Control servers communicated with the OPC UA servers of the operating environment using OPC UA binary over TCP.

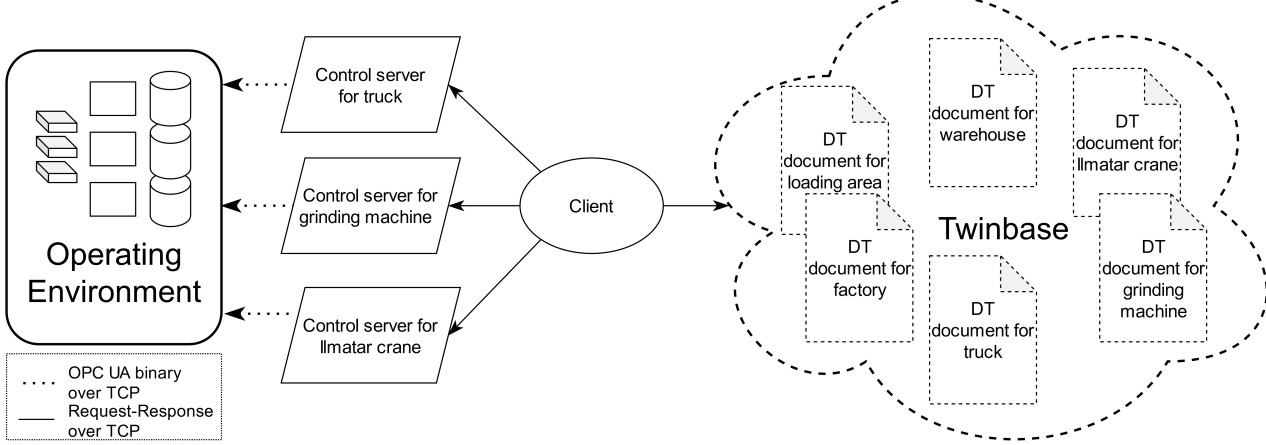

**Figure 3.** Centralized client-server control implementation corresponds to the traditional way of implementing factory communications. Twinbase use is voluntary in this architecture, and it can be integrated to be a part of the client. The arrows indicate the direction in which communication can be initiated, but data can still flow in both directions.

In the centralized client-server implementation, only one client controlled the system. The client initiates all communication between machines and the client, as shown by the arrows in Figure 3, although data flows both ways. There was no high-level embedded intelligence in machines nor in the client. Instead, the implementation relied on manual control.

In order to realize our high-level mission of moving material in the factory, the client connects to Twinbase to get the IP address of the truck from the truck's DT document. After that, the client fetches the DT document of the loading area, reads the location of the area from the document, and drives the truck to the location. Finally, the client uses the factory DT document to find all entities located in a factory. This is possible since these entities, which include both machines and non-physical objects like loading area, are stored as a child of the factory in the document. Finally, the client sends commands to the machines that are needed to process the roll.

Multiple clients control the system in the decentralized client-server implementation. Each client is responsible for executing a part of the production process by controlling machines. For example, getting an item from a warehouse, machining it, and then returning it to a different warehouse. When a certain part of the process is finished, another client is then responsible for performing the next part of the process. Clients can be spawned as needed. Decentralized client-server control implementation does not have high-level intelligence in the machines. Contrary to the centralized implementation, multiple nodes control the decentralized system, as can be seen in Figures 3 and 4. These nodes cannot communicate with each other . Instead, they can only send commands to machines and Twinbase, which then execute these commands. Besides that, this system works similarly to the centralized model.

In the decentralized P2P implementation (Figure 5), machines have intelligence, meaning a capability to execute their part in high-level mission. These intelligent machines then control the system. For example, the truck knew that it should bring the roll to the loading area and find a machine to have the roll brought to a warehouse, and the grinding machine knew to look for a new roll from the warehouse, find a machine to bring the roll to it, and after getting the roll ready, finding a machine to put the finished roll to the warehouse.

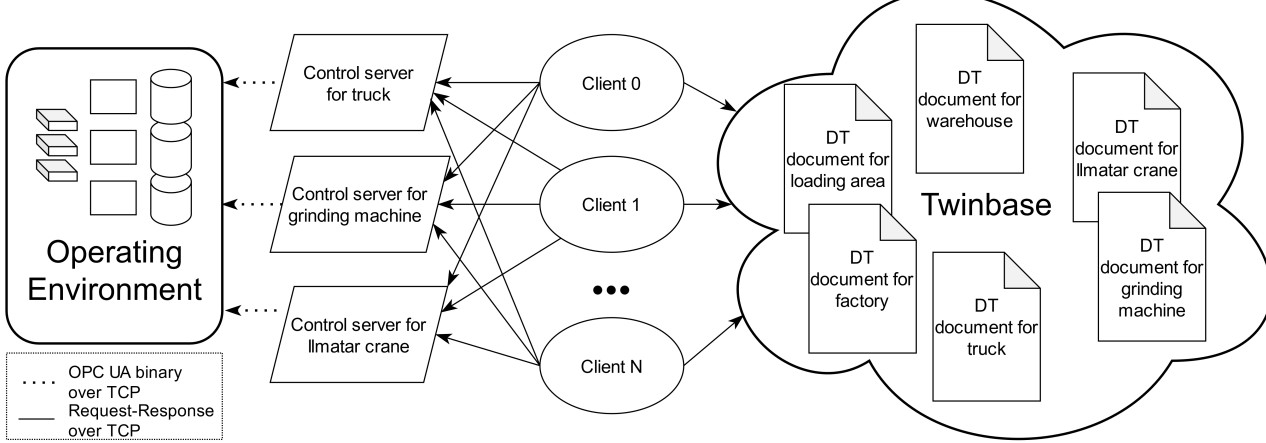

**Figure 4.** In the decentralized client-server control implementation, N number of clients together control the factory by sending commands to machines, but the clients do not communicate with each other. The arrows indicate the direction in which communication can be initiated, but data can still flow in both directions.

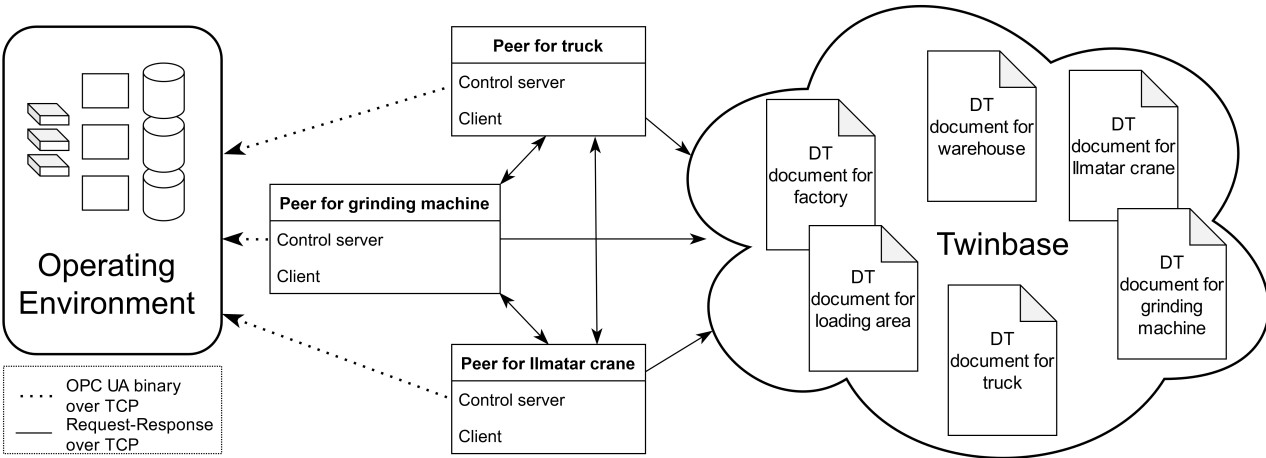

**Figure 5.** In the decentralized P2P control implementation, machines control the factory by communicating with each other. The arrows indicate the direction in which communication can be initiated, but data can still flow in both directions.

Twinbase, hosted in GitHub, was used to host DT documents. In all the implementations, the controlling unit or units requested necessary DT documents from the Twinbase for static information such as an IP address or available commands. On the other hand, the controlling unit requested dynamic information from OPC UA servers or control servers. Dynamic information contained, for example, the status of a machine or its location. Section 3.2 presents details on information available within the DT document.

## 3. Results

### 3.1. Control Implementations

The comparison of control implementation is presented in Table 1. In the centralized implementation, metadata can also be stored directly to the client. However, this makes the replacement of a client, for example, in case of maintenance, more challenging compared to storing data into Twinbase. When replacing the client while using Twinbase, a new client can fetch data from Twinbase. In the two decentralized implementations, Twinbase is always needed, or otherwise each controlling unit would need to duplicate all data stored

in Twinbase. Keeping this information up to date when machines are added, removed, replaced, or maintained, is virtually impossible.

**Table 1.** Comparison of advantages and disadvantages of different implementations. Adopted from [33].

|  | **Centralized Client-Server** | **Decentralized Client-Server** | **Decentralized P2P** |
|---|---|---|---|
| Twinbase | Beneficial | Needed | Needed |
| Control server representing warehouse | Not required | Required | Required |
| Scalability | Poor | Good | Good |
| Robust | Poor | Decent | Decent |
| Changing production | Good | Decent | Bad |
| Complex production algorithms | Client | Client | Peer (client & server) |
| Implementing production algorithm | Decent | Poor | Good |

The centralized client-server model does not require a program presenting a control server for the warehouse, unlike the two decentralized models. This is because there is only one node that can contact a warehouse and, thus, all data can be saved within the client. Although, using a control server for a warehouse in a centralized model allows easier replacing of the central client.

The centralized implementation has worse scalability than the other two implementations, as it is possible to use only one client that controls everything. In the decentralized client-server implementation, each client only controls the machines it needs for that part of the production, reducing the connections. In the decentralized P2P implementation, each node only needs to contact other nodes in the same space, reducing the connections per peer.

Because only one entity controls the centralized implementation, it is not very robust. The other two control implementations use Twinbase, a centralized system, which reduces the robustness. However, robustness can be increased by having a backup Twinbase or using cache.

Changing production is very simple in the centralized implementation, as only one client is needed to be changed. Because in the decentralized client-server implementation a new client is spawned to handle each portion of the production, current clients are not needed to be modified. In the P2P implementation, changing production is the most laborious, as it might require making changes to multiple peers.

Complex production algorithms in client-server implementations have to be implemented within the clients. This is because machines are servers, unlike in the P2P implementations where they are peers, thus cannot send any requests. Because of this, for example, when machines are moving, they cannot get the location of other machines to avoid collisions. Implementing algorithms is the most complex task in the decentralized client-server system, as clients cannot communicate with each other, and multiple clients control the system. On the other hand, implementing a production algorithm is a bit simpler in a centralized implementation, as only one entity controls the whole system and therefore needs to know the location for all the machines. Finally, implementing a production algorithm is most straightforward in the P2P implementation as each machine can control itself and get the location of other machines.

*3.2. Digital Twin Document*

Twinbase hosted DT documents in each implemented system. These documents contained various information, such as the name of the entity they represented, description of it, and its functionalities. The required information is shown in Table 2. The description

of the required fields is below. The DT documents used in this paper are listed in Table 3. These documents can be accessed by clicking the DTID (Digital Twin Identifier)

**Table 2.** Table of variables in a DT document and their descriptions.

| Variable | Description |
|---|---|
| DTID | Digital Twin Identifier: URL that redirects to a DT document. |
| IP address | Control servers IP address. |
| Location | Location for stationary objects. |
| Type | Type of the entity. For example, "Transport 3D" for the crane. |
| Parent | DTID for the parent if it exists. |
| Children | DTIDs for its children in DTW. |
| Functions | List of possible paths for control server with possible parameters. |

**Table 3.** List of names and descriptions for entities that have a DT document and links to those DT documents. DT documents used in this paper can be found using their DTIDs, which are URLs, from this table. All DT documents can be read at https://zoelz.github.io/twinbase/ (accessed on 15 February 2022).

| Name | Description DTID |
|---|---|
| Factory | Area in which all other entities are located. https://dtid.org/d1816959-8a88-40b1-9bfd-8a670b629083 (accessed on 15 February 2022) |
| Warehouse 1 | Area where rolls can be stored. https://dtid.org/59319824-39d9-423b-b6de-616047063152 (accessed on 15 February 2022) |
| Loading area 1 | Area for loading and unloading trucks. https://dtid.org/7606d2f2-2592-4073-a64b-de66c10ea585 (accessed on 15 February 2022) |
| Ilmatar Crane | Crane for moving objects in the factory. https://dtid.org/a346d686-fa08-4eab-86b7-1e5367d46e98 (accessed on 15 February 2022) |
| Truck 1 | Truck for moving objects to the factory. https://dtid.org/dde9d093-05bd-4512-8a23-1241e8809612 (accessed on 15 February 2022) |
| Grinding machine 1 | Grinding machine that is used to grind rolls. https://dtid.org/3f31bdc2-1398-497f-be7d-3386a79523a6 (accessed on 15 February 2022) |

**DTID** is used to identify and access DTs and their DT documents.

**IP address** is required to contact control servers.

**Location** for static objects. This could also be saved within objects' OPC UA servers or control servers, but this would require additional messages to get that information compared to having that information saved in Twinbase.

**Type** of entity, so that the controlling unit can choose an entity of the needed type. Type is required in automated systems, so the controlling unit knows which entity they should use and what each entity is capable of doing.

DTIDs for **parent** and **children** DTs. Listing parent DTs allows finding the environment in which the machine operates. The parent's children can be used to fetch other entities, such as machines, in the same environment. Children of a machine can also represent its subcomponents.

**Functions** provided by DT. This allows the controlling unit to find functions it can call and their parameters. Functions should have standardized names and functionality. Similar machines should have the same functions with the same parameters so that when a system is automatized, the controlling entity only needs to find a machine with the right type and check that the correct function exists for it.

*3.3. Twinbase in Practice*

Figure 6 demonstrates the usages of Twinbase in practice. The first client requests the DT document for the loading area from Twinbase. After that, the client gets the loading area's location from DT documents and sends this as a target value for the truck's drive function. After the truck arrives at the loading area client requests from Twinbase the DT document for the parent of the loading area using the parent's DTID in the loading area's DT document. Using this parent's DT document, which is the factory, the client gets all of its childrens' DTIDs and requests DT documents for them. From these DT documents, the client looks for a transport type of machine, and when it finds one, it commands it to drive to the loading area and pick up the roll from the truck. In this description, it was assumed that the client knew the information from the DT document of the truck beforehand.

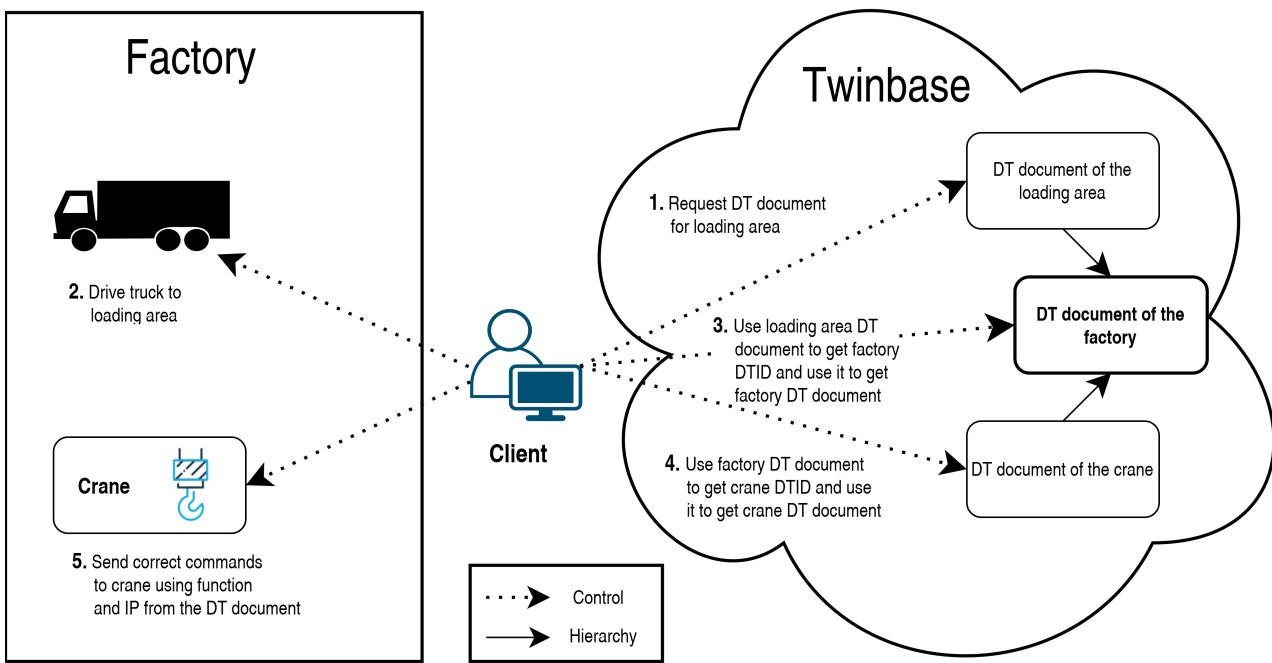

**Figure 6.** Practical example of applying Twinbase to find entities in the same operating area.

## 4. Discussion

This paper used DT documents to enable the control of machines in a smart factory, and three types of control methods, P2P, centralized client-server, and decentralized client-server, were investigated more carefully. To examine these control methods, we created a proof-of-concept simulation model using ROS and Gazebo. The simulation model and qualitative analysis showed that the P2P control model offers the most flexible way of implementing control in a smart factory. Flexibility is seen as one of the main goals for the production management in Industry 4.0. The P2P control model allows each machine to communicate with all the other machines and allows self-organization of production. This paper opens discussion on how control of machines could benefit from digital twins and how machines should be controlled in a smart factory.

### 4.1. Communication Methods in Smart Factory

Each machine in the simulation model had an OPC UA server that mimicked the functionalities of the real OPC UA server of the machine. Therefore, by only changing the IP addresses of the OPC UA servers, the developed control methods can be applied to a real factory. OPC UA servers are becoming the standard solution to communicate with machines and their PLC systems: An OPC UA server allows reading the internal sensor values of the machine, and sending control commands to it. In this paper, a control server that accepted HTTP requests was implemented on top of each OPC UA server. This enables interoperability with almost any device since HTTP is the most widely used Internet protocol. Based on the simulation model and feature comparison of different control models, the distinguishing factors of the control methods are flexibility, scalability, and where the production algorithms are implemented. We also noted that implementing production algorithm and self-adapting systems is easier with the P2P control model.

### 4.2. DT Document as an Enabler for Control in Smart Factory

This paper follows the information-oriented view on digital twins [15]. In this view, the most important functionality of a digital twin is to make the information from the real-world entity available. To make information available, the information sources and digital twin metadata are described with a DT document. The goal of the DT document is to provide information on how to access all information of the machine. This information also includes the communication interfaces and the supported communication methods.

In this paper, the DT document was the central element that allowed communication between the machines. The machines used the DT document to find the variables of other machines shown in Table 2. These variables contain information, such as IP address, location, type and relations between machines. Use of DT documents ensures that the most recent information is available to all entities. The only control model that does not necessarily need DT documents is the centralized client-server model. In this method, the information of the machines is stored to a centralized database of the centralized control unit.

The need to describe a machine and its capabilities, is also recognized by large companies and associations. Examples of standardization efforts towards unified digital twin description include: World Wide Web Consortium Web of Things Thing Description, Microsoft Digital Twin Definition Language, and Asset Administration Shell. In this paper, the DT document, first introduced in our previous paper [19], was used to describe the digital twin. Nevertheless, we are carefully following the development of the other standards, and may adopt these standards in the future.

### 4.3. Future Work

The next phase in the development of DTs is the standardization of their descriptions using DT documents. In the best-case scenario, there is only one dominating DT standard in the future. Another major step in the development of DTs will be the network of DTs. DTW is required to make digital twins discoverable and accessible. In the future, similar search engines such as Google, could also be developed on top of DTW.

Finally, the simulation model should be modified to represent a whole smart factory with a real production process. Currently, the model only represents an artificial subtask of a larger production process in which a roll is ground. Based our preliminary tests our operating environment can be scaled and also be used simulating larger tasks, but if it is not able to be scaled it can be replaced for instance with the Rviz as it is a visualization tool, unlike Gazebo, which has a physics engine included. Although, the visualization of the simulation could be improved with Unity, which we are currently researching. After improving the simulation model, measurements could be performed using the model to support the qualitative analysis of control methods in this paper.

## 5. Conclusions

This paper investigated different methods to implement the control of machines using DT documents and their distribution service, Twinbase. A DT document provides up-to-date description of the features and interfaces of cyber-physical entities, e.g., machines, to all stakeholders in a smart factory. Via these interfaces, the cyber-physical entities can communicate with each other. For the investigation of control methods, a ROS and Gazebo based proof-of-concept simulation model of a smart factory was created. This model is freely available on Github (https://github.com/Zoelz/simulationModelControl, accessed on 15 February 2022). The proof-of-concept simulation showed that P2P communication allows maximal flexibility when implementing communication between machines.

The paper shows that creating a network of DTs—called Digital Twin Web or DTW—is beneficial for the creation of self-adapting smart factories. DTW allows distribution and discoverability of DT documents analogously to the early World Wide Web that allowed accessing (hyper-)text documents. Storing DT information in an easily accessible form is a necessity for the wide adoption of digital twins in smart factories and enables the control of machines. Future work includes scaling the simulation model up and continuing the standardization efforts of the unified description of digital twins.

**Author Contributions:** Conceptualization, J.M., R.A.-L., and J.A.; methodology, J.M.; software, J.M.; validation, J.M., R.A.-L., and J.A.; investigation, J.M.; resources, K.T.; writing—original draft preparation, J.M.; writing—review and editing, J.M., R.A., J.A., P.S., and K.T.; visualization, J.M., P.S., and J.A.; supervision, K.T.; project administration, P.S.; funding acquisition, K.T. All authors have read and agreed to the published version of the manuscript.

**Funding:** This research was funded by the Business Finland under Grant 3508/31/2019 and ITEA 3 Call 5 MACHINAIDE.

**Data Availability Statement:** Not applicable.

**Conflicts of Interest:** The authors declare no conflict of interest.

## Abbreviations

The following abbreviations are used in this manuscript:

| | |
|---|---|
| DT | Digital Twin |
| DT document | Digital Twin document |
| DTID | Digital Twin Identifier |
| DTW | Digital Twin Web |
| ROS | Robot Operating System |
| P2P | Peer-To-Peer |

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
