# Peer review of "Using Digital Twin Documents to Control a Smart Factory: Simulation Approach with ROS, Gazebo, and Twinbase"

_machines, doi:10.3390/machines10040225_

Round 1

Reviewer 1 Report

The paper gives a simulation approach by using digital twin documents to control smart factory, and different control methods have been compared. The questions are given as follows:

1. What is the difference between the Twinbase and cloud? It seems like a traditional cloud.

2. The simulation scene in this paper is very simple, and how to handle a real smart factory with much more machines?

3. The conclusion part is too long, effective summary should be given, and there is a repetition.

Author Response

Dear reviewer,

Thank you for your comments. The detailed response to your concern is below:

Concern 1: " What is the difference between the Twinbase and cloud? It seems like a traditional cloud."

Authors response: The authors would like to thank the reviewer for the comment, as the difference between cloud and Twinbase might be unclear to readers. Twinbase is a traditional static web server with special features for digital twin documents and the ability to be hosted from GitHub Pages. Readers who are more interested in Twinbase can refer to Autiosalo et al. (2021) paper that is ref. [24] in the manuscript.

Authors action: We added further clarification about Twinbase to the introduction.

Concern 2: "The simulation scene in this paper is very simple, and how to handle a real smart factory with much more machines?"

Authors response: The authors would like to thank for the comment. The simulation model does not try to present a whole factory but a part of a factory. However, based on our preliminary tests, Gazebo seems to scale enough that it could be used to simulate a whole factory, and if it does not, it could be replaced with a program that does not have a physics engine like Rviz.

Authors action: Authors added mention about the scalability of the model to the discussion chapter.

Concern 3: "The conclusion part is too long, effective summary should be given, and there is a repetition."

Authors response: The authors would like to thank for the comment. We agree that the conclusion is too long and could be shortened.

Authors action: We shortened the conclusion section and removed the repetition.

Reviewer 2 Report

  1. What is the difference between normal M2M communication and DT-documents-based M2M communication? Please make it clear in the Introduction.
  2. The gap of the proposed method is not clear in the Introduction. Although this paper provides the implemented simulation environment in open source platform, simple implementation of DT does not give contribution to research area. The ROS, Gazebo, Python, and OPC UA are just well-known contents for implementation of DT.
  3. Please compare the proposed method with the methods in existing literature.
  4. The research terms used in this paper need to be revised by referring to other studies.

Author Response

Dear reviewer,

Thank you for your comments. The detailed response to your concern is below:

Concern 1: "What is the difference between normal M2M communication and DT-documents-based M2M communication? Please make it clear in the Introduction."

Authors response: The authors would like to thank for the comment. In DT-document-based M2M communication, there is an added metadata layer. This metadata layer allows automated and more scalable communication as all information about the machines can be read from the DT documents located in Twinbase.

Authors action: Authors added an explanation of differences to the introduction.

Concern 2: "The gap of the proposed method is not clear in the Introduction. Although this paper provides the implemented simulation environment in open source platform, simple implementation of DT does not give contribution to research area. The ROS, Gazebo, Python, and OPC UA are just well-known contents for implementation of DT."

Authors response: The authors would like to thank for the comment. The research gap is to allow a vendor-lock-free and distributed way of storing information that allows M2M communication. In addition, this enables self-organizing M2M communication within a factory.

Authors action: Authors added the research gap to the introduction.

Concern 3: "Please compare the proposed method with the methods in existing literature."

Authors response: The authors would like to thank for the comment.

Authors action: The authors added a paragraph in introduction presenting ways how machines can find other machines currently.

Concern 4: "The research terms used in this paper need to be revised by referring to other studies."

Authors response: The authors would like to thank for the comment.

Authors action: The authors added reference to other studies for terms in places where it seemed relevant.

Reviewer 3 Report

This paper deals with digital twin documents and their usage for machine-to-machine communication.

The entire approach is relatively well described and seems to be sound.

The structure of the paper is relatively good. However, the state-of-the-art part, which is now part of Section 1 should be moved to a new Section 2. It is better to have a separate state-of-the-art section than to have it merged with the introduction, because the introduction should be relatively short to give the reader the information, what is described in the paper and why.

The references in the paper seem to be relevant and up to date.

The figures are appropriate and of sufficient printing quality.

The English is very good, he amount of typos and errors is quite low. Proofreading by a grammar-skilled native speaker can still prove useful, though.

Author Response

Dear reviewer,

Thank you for your comment on the paper structure. We agree that the introduction part of the paper is rather long. However, the introduction only contains essential information. We think that it is necessary to provide a comprehensive description of the background information, including smart factories and digital twins. In addition, we think that the research gap should be mentioned in the introduction section, and the paper flow is better if the necessary information is already provided in the introduction. The existing literature on digital twin enabled machine-to-machine communication is scarce, and it is not justified for that reason to create a separate section for the state-of-the-art review.

Round 2

Reviewer 2 Report

Clear revision is provided by authors. I decide to accept this paper for publication.